# Appraising the quality standard of clinical practice guidelines related to central venous catheter-related thrombosis prevention: a systematic review of clinical practice guidelines

Jing Zhang, Yongya Wu, Shuai Zhang, Wenmo Yao, Faqian Bu, Aoxue Wang, Xiuying Hu [ORCID], Guan Wang [ORCID]

Innovation Center of Nursing Research, Nursing Key Laboratory of Sichuan Province, State Key Laboratory of Biotherapy and Cancer Center, West China Hospital, Sichuan University /West China School of Nursing, Sichuan University, Chengdu, China

**Correspondence to**
Guan Wang;
guan8079@163.com and
Dr Xiuying Hu;
huxiuying@scu.edu.cn

## ABSTRACT

**Objective** To evaluate the quality and analyse the content of clinical practice guidelines regarding central venous catheter-related thrombosis (CRT) to provide evidence for formulating an evidence-based practice protocol and a risk assessment scale to prevent it.

**Design** Scoring and analysis of the guidelines using the AGREE II and AGREE REX scales.

**Data sources** Pubmed, Embase, Cochrane Library, Web of Science, CNKI, Wanfang, VIP, and the Chinese Biomedical Literature, and the relevant websites of the guideline, were searched from 1 January 2017 to 26 March 2022.

**Eligibility criteria** Guidelines covering CRT treatment, prevention, or management were included from 1 January 2017 to 26 March 2022.

**Data extraction and synthesis** Three independent reviewers systematically trained in using the AGREE II and AGREE REX scales were selected to evaluate these guidelines.

**Results** Nine guidelines were included, and the quality grade results showed that three were at A-level and six were at B-level. The included guidelines mainly recommended the prevention measure of central venous CRT from three aspects: risk screening, prevention strategies, and knowledge training, with a total of 22 suggestions being recommended.

**Conclusion** The overall quality of the guidelines is high, but there are few preventive measures for central venous CRT involved in the guidelines. All preventive measures have yet to be systematically integrated and evaluated, and no risk assessment scale dedicated to this field has been recommended. Therefore, developing an evidence-based practice protocol and a risk assessment scale to prevent it is urgent.

## STRENGTHS AND LIMITATIONS OF THIS STUDY

⇒ Two quality evaluation tools, AGREE II and AGREE REX, were used to comprehensively and rigorously evaluate the quality of central venous catheter (CVC)-related clinical guidelines.

⇒ This review clarifies the CVC guidelines with high evidence strength and the highest clinical applicability and provides a reference for clinical medical personnel to select the best guidelines.

⇒ The evidence in the guidelines included in this study is mainly from developed countries and some developing countries, and other countries need to be cautious when interpreting the results of this study.

⇒ Only guidance documents in English and Chinese were included, so we cannot rule out the possibility that some guidelines may have been omitted from this study.

## INTRODUCTION

The central venous catheter (CVC) is inserted through the proximal central or peripheral vein, with the catheter tip located at one-third of the junction between the superior vena cava and the right atrium.[1] The CVC includes subclavian vein/jugular vein/ femoral vein puncture catheters, peripherally inserted central catheter, and implantable venous access port.[2] It has many advantages, such as convenient and safe operation, rapid establishment of intravenous access, reduced puncture times, and infusion of irritating liquid; therefore, it has been widely used in chemotherapy, haemodialysis, nutrition support, and surgery support.[3–6]

Catheter-related thrombosis[7] (CRT) refers to the formation of blood clots on the outer wall or inner wall of a catheter. CRT is one of the common complications after catheterisation of CVC, which can lead to unplanned extubation and pulmonary embolism, affecting the progress of treatment and increasing the cost of treatment.[8–11] The risk factors of CRT could be categorised into patient-related, disease/treatment-related, and catheter-related factors. Patient-related factors include sex, advanced age, long-term bedridden, venous thromboembolism

(VTE) history, and acquired or hereditary blood hypercoagulability state.[12–14] Diseases/treatment-related factors include the grade of cancer, chemotherapy, radiotherapy, parenteral nutrition support, surgical trauma, or excessive bleeding.[15] Catheter-related factors include catheter material, catheter-to-vein ratio, puncture times, catheter indwelling length, and the catheter tip's location.[16]

CRT has the characteristics of high incidence, early onset, and clinically silent symptoms.[17] The clinical manifestations of CRT are as follows: deep venous thrombosis,[18] superficial thrombophlebitis, asymptomatic thrombosis, and thrombotic catheter failure. The study reported that CRT incidence ranged from 1.9% to 73.0%.[5 9 16 17 19] It could happen as early as the first day after catheterisation,[17 20] and the average time of occurrence was within 2 weeks after catheterisation.[5 11 21 22] However, the early symptoms and signs were silent; only a few patients showed clinical symptoms. Due to the limited medical resources, vascular surgery in most medical institutions in China cannot monitor the occurrence of CRT in time. It cannot correctly prevent and deal with CRT, increasing hospitalisation time, resulting in irreversible effects, and affecting the quality of follow-up treatment of patients.

Therefore, assessing and taking measurements early to prevent patients at high risk of CRT are essential to ensure clinical workers formulate their safety. Clinical practice guidelines after systematically integrating and evaluating relevant research and clinical experience according to evidence-based concepts to a particular common symptom, disease, or problem, to provide guiding suggestions for clinical workers and patients, and scientifically promote the application of evidence in clinical practice.[23] Several clinical practice guidelines about CRT prevention have been published, but their quality could be more apparent. Therefore, through the systematic retrieval, screening, and reading of clinical practice guidelines related to CRT prevention and the quality evaluation and content analysis of the guidelines, this review was expected to provide a scientific basis for developing evidence-based practice protocols and risk assessment scales for CRT.

## INFORMATION AND METHODS
### Search strategy
#### Search databases
We searched English and Chinese databases, including Pubmed, Embase, Cochrane Library, Web of Science, CNKI, Wanfang, VIP, and the Chinese Biomedical Literature Database from 1 January 2017 to 26 March 2022. In addition, the relevant websites of the guideline, including the National Institute for Health and Care Excellence (NICE) and Medline, were searched. A manual search of relevant studies and bibliographies was also adopted.

#### Search strategies
The literature retrieval adopted a strategy of the combination of MeSH terms and free words. The MeSH terms and synonyms of CRT and clinical practice guidelines were searched in English and Chinese databases. Take Pubmed as the retrieval representative of the English database. The search strategy is shown in Supplementary Materials—Search strategies.

### Inclusion and exclusion criteria
The inclusion criteria were (1) guidelines including the contents of treatment, prevention, or management of CRT; (2) guidelines published or updated after 1 January 2017; and (3) guidelines published in English or Chinese. The exclusion criteria were (1) guidelines developed by expert consensus exclusively, (2) incomplete guidelines that are in the process of development, and (3) the translation or reinterpreting of the guidelines.

### Guideline screening and data extraction
Two researchers who had received systematic literature retrieval and training searched the literature. The searched literature was imported into Endnote X9 software, and the duplicates were removed. Then, two researchers independently read the title and abstracts for preliminary screening to exclude unrelated articles. Subsequently, two reviewers read the full text carefully and screened it again according to the established criteria for inclusion and exclusion. In case of disagreement, the third party will be consulted before determining. Finally, a predefined data extraction table was used to extract the data from the included studies.

### Quality appraisal
The AGREE II scale,[24] a widely used tool for evaluating the quality of the guideline, was used to evaluate the included guidelines. It includes six domains, that is, scope and purpose, rigorism, independence, clarity, application, and independence of writing, with 23 items. Each item has a maximum score of seven and a minimum of one. Seven indicates that the evaluator is fully satisfied with the area covered by the entry, and one indicates that the evaluator is entirely dissatisfied with the field. The higher the evaluator is satisfied with the area covered by the guideline, the higher the scores. The evaluator should score the satisfaction degree of the guideline according to the actual situation to ensure the fairness of the guideline. Three researchers who systematically trained the use methods of the AGREE II scale were selected to evaluate the guidelines to ensure that the researchers had a complete understanding and consistent understanding of the items and to reduce the biases of evaluation. If there is a dispute over an item, experts in the relevant field would be consulted to resolve it. The standard rate of the three researchers in each field of the guideline was calculated, and the systematic evaluation content analysis and summary of the guideline were carried out based on the evidence-based concept. Level A recommended the guideline, indicating that most entries in the guideline scored ≥5, and its standardised percentage was ≥60%. Level B was recommended after modification and

indicated that the standard rate in more than three areas was 30%–60%. Level C indicated that the guideline was not recommended and that the standardised rate in three or more areas was less than 30%.

To evaluate the quality and applicability of each recommendation, improve the comparability of the results, and increase the effectiveness of the evaluation process, the AGREE REX scale[25] was used to evaluate the included guidelines. Three researchers systematically trained using the AGREE REX scale and evaluated the guidelines. The scores of the three researchers in each area of the guidelines were standardised to calculate. The guidelines were systematically evaluated, and content was analysed and summarised based on the evidence-based concept. Referring to the recommendations of the AGREE REX manual, the guidelines define an overall score of >70% as high quality, an overall score of <30% as low quality, and others as moderate quality.

### Statistical methods
SPSS V.22.0 was used to count the scores of the three researchers in each field of the guideline, and the Intraclass Correlation Coefficient (ICC) of the data was calculated. According to the results, the consistency of the two researchers' guideline evaluation was tested. ICC<0.4 indicated poor consistency, 0.40≤0.59 indicated that the consistency was general, 0.60≤0.74 indicated good consistency, and >0.74 indicated perfect consistency.

### Patient and public involvement
No patients or members of the public were involved in the development of the research question, the design or the conduct of the study, or the interpretation and writing up of results.

### RESULT
### Guideline retrieval results
A total of 2092 articles were retrieved, and 571 were removed by software and manual deletion. After reading the title and abstract, 1142 articles were excluded; 379 articles were screened according to inclusion and exclusion criteria; 267 articles that did not meet the article's theme were excluded, five related guidelines were traced through the snowball strategy, and 117 articles were finally selected; 11 articles were not obtained, and 106 complete articles were obtained. The full-text screening excluded 18 non-evidence-based guidelines issued in individual form, six guidelines that had been updated, and 73 guidelines that did not include CRT treatment, prevention, and management. Finally, nine literature were selected, including two Chinese literature and seven English literature. The specific screened process is shown in online supplemental figure S1.

### Essential characteristics of inclusion in the guideline
The nine guidelines were from the American Society of Clinical Oncology, the International Initiative on Thrombosis and Cancer, the Chinese Medical Association, the Spanish Medical Oncology Society, the Infusion Nursing Association, and other international authoritative medical associations. Only one guideline targets CRT risk assessment, prevention, and intervention.[20] Five guidelines were specifically for venous thrombosis prevention in cancer patients,[15 26–29] and the content related to CRT was only part of it. Two are guidelines for ordinary patients' venous thrombosis.[26 30] CRT patients are part of the content, and only one guideline targets clinical infusion therapy.[30] CRT is part of the treatment (see online supplemental table S1).

### Results of AGREE II and AGREE REX
The included guidelines used AGREE II and AGREE-REX to evaluate the quality of methodologies and recommendations, respectively. After scoring the nine guidelines on the AGREE II scale, it was found that three guidelines[26 31 32] were recommended as A, and six guidelines[27 29 33–36] were recommended as B, as shown in online supplemental table S2. The average score in all areas was more than 60% excellent. The ICC of the scores of the three reviewers for each domain of each guideline was more significant than 0.8, indicating that the agreement among the reviewers was good. Specifically, area 1 ('Scope and Purpose') addresses the general scope and purpose of CPGs (clinical practice guidelines), the specific clinical problems that need to be addressed, and the target population. The average score was 66.67%, of which eight articles scored more than 60% and five papers scored more than 70%. Area 2 ('Participants') reflects whether the normative process includes input and participation from relevant stakeholders. The average score was 60.70%, and six of the articles scored above the average score. Regarding 'rigour' (area 3), the average score was 61.88%, with seven guidelines scoring above the average. Guideline 4[35] has the lowest scores in the three areas of scope and purpose, who is involved, and rigour, while most of the rest describe the development process, although with varying levels of detail.

Area 4 ('Clarity') focuses on the specificity and clarity of these items, including the clear articulation of options for health management and making critical recommendations that are easy to identify. With an average score of 64.81%, there was the slightest difference in all guideline scores in this area, even though guideline scores 1, 2, 8, and 9 were below the average. 'Application' (area 5) refers to facilitators and obstacles to implementing recommendations within the context of the guidelines, strategies for their implementation, and possible resource implications. The CPGS scored an average of 60.49% in this area, with three scores below 60%. Finally, the sixth area ('independence') aims to ensure a lack of bias in developing guidelines. The score in this area was relatively good, with a median CPGS of 63.58%, three above 70%, and seven above 60%: most described funding, potential impact, or associated conflicts of interest. The guidelines scored slightly worse than the others in the areas

of participants, rigour of development, applicability, and editorial independence.

After evaluating the nine guidelines according to the AGREE REX scale, the overall scores of guidelines 3[34] and 6[32] were >70%, which was of high quality, and the overall scores of the other guidelines were ≥30%, which was of moderate quality (see online supplemental table S2). The three reviewers agreed well, with an ICC value greater than 0.8, and each guideline scored slightly lower than the implement ability domain in the values and preference domains and clinical practicality. In clinical applicability, guideline 6[32] scored the highest, 83.33%, with a large score difference between guidelines, guideline 1[33] had the lowest score of 42.59%, as detailed in online supplemental table S2.

### Comparative analysis of the specific content of CRT preventive measures in the guidelines

After reading all the recommendations and precautions included in the guidelines, it was found that the content of CRT prevention can be mainly divided into three aspects: risk assessment, preventive measures, and health education. The specific content and methods are shown in online supplemental table S3. Due to differences in the topic of the guidelines and the study participants, the focus and level of detail of the prevention strategies recommended for CRT vary among the guidelines. Therefore, for analysis and comparison, the relevant contents are summarised in online supplemental table S4.

## DISCUSSION
### Essential characteristics of CRT prevention guidelines

The guidelines included in the study are widely used and have high impact factors. Guidelines issued by the China Branch of the International Union of Angiology, for the first time, the preventive measures of CRT were systematically summarised, which filled the gap in this field. However, the methodological quality could have been better. Guidelines were issued by ASCO, SEOM, ASH, NCCN, and ITAC-CME.[15 26 27 32] This paper mainly expounds on the preventive measures of CRT from two aspects of drug prevention and CVC-related prevention, including the choice of anticoagulant drugs, the time of continuous anticoagulant treatment, and the choice of catheter location; however, these were only a tiny part of the guideline, and the content was scattered in the included guidelines. Five included guidelines were targeted at cancer patients and had certain limitations. Guidelines issued by the Chinese Medical Association and NICE[26 30] mainly put forward some suggestions on how to prevent various types of thrombosis, and the prevention measures for CRT were one of them, including drug prevention of CRT, CVC catheter-related prevention, and physical prevention. The guidelines issued by INS describe the latest clinical infusion treatment practice standards and specific measures.[30] The prevention of CVC-related complications was part of them, mainly describing how to deal with CRT.

### Quality evaluation of the guidelines

To ensure the quality of the guidelines, researchers should follow the principles of professionalism, fairness, evidence-based, and transparency when developing the guidelines. As the guidelines included in this study are of good quality and high impact factors, the AGREE II scale is used to evaluate the six areas of the guideline. The guidelines' clarity, scope, and purpose have reached the standard level. In contrast, the rigorism and independence, participants, and application of the guidelines must be further improved. The specific analysis was as follows: rigorism and independence.

Regarding rigorism, the expert consensus was released by the China Branch of the International Union of Angiology. The guideline does not systematically retrieve CRT prevention-related research but only summarises the 'experience' of clinical experts, lack of corresponding evidence support, and lack of scientific and rigorous. The guidelines issued by NCCN and NICE,[26 29]29 February 2024 1:23:00 PM systematically evaluated the relevant research and conclusions and recommended opinions and suggestions. However, these two guidelines should have pointed out the steps and methods of systematic retrieval of evidence and the criteria for accepting and excluding the research. In addition to the guidelines issued by INS,[30] the rest do not specify the steps and methods for updating the guidelines. However, the guidelines indicated the timing of the update[30] and the extent to which the evidence level of the new guideline should be higher than that of the previous edition. They indicated that the quality of the evidence in the guideline is constantly improving. Due to the differences in medical and health systems at home and abroad and the nonstandard evaluation of guidelines, the study found that most of the guidelines published abroad put forward statements of interest in the study.[15 26 27 29 30 32] However, the guideline issued by the China Branch of the International Union of Angiology failed to state the study's statement of interest. The interests of the panel members with other bodies are also not recorded, and the fairness and independence of the study are not guaranteed. Therefore, we should pay full attention to the rigour and independence of the guideline, pursue evidence according to the concept of evidence-based medicine, and avoid the influence of external factors and research biases caused by personal preferences to ensure the rigour and independence of research.

### Participants

In the participant section, only two guidelines issued by the Chinese Medical Association and INS,[27 30] the names and work units of all the experts involved in the formulation of the guidelines are listed, while the rest of the guidelines refer only to interdisciplinary teams or association support and without specify the relevant

information of the researchers. In addition to the release guidelines of the Chinese Medical Association, NCCN, and ITAC-CME,[23 25 30] the rest of the team did not explain how to collect comments and suggestions from users considering the relevant information of researchers, such as work experience, units, and professional titles, can reduce the bias caused by personal factors of researchers, which is conducive to the formulation of guidelines and reflects the universality and authority of the formulation of guidelines.

## Application

Relevant studies have shown that the use of clinical guidelines could be much higher in the field of application. Most of the guidelines in this study need to fully consider the difficulties and obstacles encountered in the follow-up application and the relevant systems and documents accompanying them. There needs to be training for clinical staff in using the guidelines, which is not conducive to learning and using the guidelines. Of the included guidelines, only this issued by NICE is presented.[30] The guidelines are applied to specific clinical methods and measures. However, the rest of the guidelines must address the forward and reverse factors in applying the guidelines, how to use potential resources, and how to transform the theory into practice. Therefore, it is necessary to carry out pre-experiments in the process of guideline formulation in the future, check the feasibility of guideline recommendations in advance, collect the problems in applying guidelines in advance, make corresponding preparations, and promote the transformation of guidelines to clinical practice.

## Analysis of the recommendations in the guidelines risk screening

Timely assessment of risk factors for CRT formation in patients is beneficial to the clinical prevention of CRT formation. In addition to the guidelines issued by SEOM and ITAC-AEM,[15 27] the rest indicate that CRT risk factors must be assessed before CVC intubation. The guidelines also clearly point out the high-risk factors of CRT, mainly divided into patient-related, catheter-related, and disease/treatment-related factors. The patient-related factors were age, sex, family history or personal history of venous thrombosis, hypercoagulable state of blood, and abnormal coagulation gene. Catheter-related factors were catheter material, placement technique, location and type of catheter, and puncture times. Disease/treatment-related factors were antiangiogenic therapy, hormone therapy, parenteral nutrition, radiotherapy, tumour and trauma, and complicated surgery. However, there is yet to be a recognised scale dedicated to CRT risk assessment. Most of the guidelines recommend using the VTE risk factor assessment scale, and the two guidelines recommend using the Khorana scale and Wells scale.[22 23 30] The guideline issued by NCCN recommends that patients with medical diseases can use the Padua scale.[29] At the same time, it is suggested that myeloma patients should

be treated with targeted anticoagulant therapy after scoring with IMPEDE VTE(Immunomodulatory agent; Body Mass Index; Pelvic, hip or femur fracture; Erythropoietin stimulating agent; Dexamethasone/Doxorubicin; Asian Ethnicity/Race; VTE history; Tunneled line/central venous catheter; Existing thromboprophylaxis) and SAVE(The Speech Arm Vision Eyes) scales. CRT risk assessment should be conducted at admission, before and after catheterisation and replacement, and when symptoms and signs occur. Timely evaluation of CRT-related symptoms of patients, regular examination of related physiological and biochemical indicators according to the actual situation, flexible adjustment of anticoagulant programme, and the timely report of the occurrence of particular circumstances.

## Prevention strategies

Through the summary and analysis of the nine guidelines, it is found that the preventive measures are mainly divided into the following five parts: classified prevention, physical prevention, drug prevention, standard CRT prevention measures, and multidisciplinary prevention. Classified prevention emphasises individualised prevention according to the patients' factors and the clinical manifestations of CRT. Physical prevention is mainly to help patients exercise actively or passively, promote their blood circulation, and reduce thrombosis formation. To urge patients to get out of bed as soon as possible, put tube side limbs to perform functional exercises such as repeated clenching or loosening, promote blood circulation, replenish water, and prevent dehydration and blood coagulation. However, most guidelines indicate that patients at risk of post-thrombosis syndrome do not recommend routine use of elastic socks for prevention, as they may aggravate skin breakage and injury. Drug prevention mainly uses low-dose unfractionated heparin, low molecular weight heparin, and vitamin K antagonists, while some guidelines recommend the use of a new direct oral anticoagulant. The risk of bleeding should be considered when anticoagulants are used, and the risks and benefits of the drug should be assessed and used with the patient's informed consent. The common prevention of CRT is mainly the treatment of a catheter, including the choice of catheter diameter and lumen, the way of catheterisation, the location of catheterisation, the time of catheterisation, and subsequent anticoagulant therapy, as shown in online supplemental tables 4 and 5. Multidisciplinary prevention refers to constructing a multidisciplinary team composed of vascular surgery, oncology, evidence-based medicine centres, medical laboratory technicians, and nurses to carry out CRT prevention to avoid the limitations of clinical practice.

## Knowledge training

The guideline issued by INS suggests that nurses should constantly update their clinical knowledge and skills, and based on the evidence-based transformation of the latest research into high-quality

evidence, to promote its transformation to clinical practice. Currently, the update of clinical nursing is slow, and most medical staff are still based on traditional nursing methods, so it is not easy to introduce new research results into clinical practice. Therefore, it is necessary to carry out systematic scientific research education and organise reading exchange meetings and knowledge training for nurses. The guidelines issued by ASCO[26] indicate that, although CRT is closely related to cancer, patients' awareness of CRT's risks and warning signals is still deficient, emphasising the need to increase patient education and awareness.[28 37] At the same time, a study suggests that patients may not report new symptoms unless asked directly, as they usually consider symptoms to be a manifestation of cancer or a side effect of treatment. Therefore, health education for patients is helpful to ensure effective communication between medical staff and patients and is conducive to patients' understanding of the disease.[38]

### Limitations of the guidelines

Although one of the nine guidelines in this study is a specific guideline on CRT prevention, it needs to retrieve relevant research and evidence systematically. It only recommends and compiles guidelines based on the experience of clinical experts, which need more scientific. Most other guidelines are suitable for cancer patients, and the primary prevention strategies related to CRT are drug prevention and CVC catheter prevention. However, their contents are fewer, their views are scattered, the subjects of the guidelines are limited, and there needs to be a more systematic, integrated evaluation. Therefore, the formulation of evidence-based CRT prevention-related clinical practice guidelines is helpful to guideline clinical staff in standardising the operation and ensuring the treatment quality of patients. At the same time, after analysing the guideline's content, it is found that no tool is dedicated to CRT risk assessment. Presently, the guidelines recommend using a mature VTE risk assessment scale. However, this kind of scale does not involve the evaluation of the CVC catheter, which is not targeted, which may lead to a measurement bias. Therefore, developing a risk assessment scale for CRT is conducive to accurately selecting appropriate preventive measures and anticoagulant treatment methods and reduces the injury caused to patients after CRT.

In summary, the quality of the nine clinical guidelines included in this study is moderately high, but there are some differences in the guidelines in different regions and countries. The guidelines need to be further improved in the following aspects: (1) there is no risk assessment scale dedicated to CRT, and a targeted evaluation form should be developed based on evidence-based evidence, which is conducive to the accurate selection of appropriate preventive measures and anticoagulation treatments and (2) clinical practice guidelines and risk assessment scales

related to CRT prevention based on evidence-based concepts should be formulated to enable patients to participate in the study based on the clinical status quo to increase the practicability of the guidelines.

## CONCLUSION

The quality of the nine guidelines is good, and the corresponding advice and suggestions about CRT prevention are provided. However, the guidelines still need to be improved regarding the rigour and independence of writing, participants, and applications. At present, there is still a need for high-quality clinical practice guidelines for CRT prevention, and there is also a need for dedicated CRT risk assessment tools. Therefore, there is an urgent need to develop evidence-based CRT prevention-related clinical practice guidelines and risk assessment tools based on the clinical situation to enable patients to participate in the management of CRT and, after that, to better manage CRT.

**Contributors** GW and XH conceived and guaranteed the project. JZ, YW, and SZ summed up the literature and drafted the manuscript. WY, FB, and AW drew the figures and tables. GW and JZ revised the manuscript. All authors contributed, edited, and approved the final manuscript.

**Funding** This work was supported by the Sichuan Science and Technology Program (Grant 2022NSFSC1290) and the West China Nursing Discipline Development Special Fund Project, Sichuan University (Grant HXHL21011).

**Competing interests** None declared.

**Patient and public involvement** Patients and/or the public were not involved in the design, or conduct, or reporting, or dissemination plans of this research.

**Patient consent for publication** Not applicable.

**Ethics approval** Not applicable.

**Provenance and peer review** Not commissioned; externally peer reviewed.

**Data availability statement** No data are available.

**ORCID iDs**
Xiuying Hu http://orcid.org/0000-0003-1470-4108
Guan Wang http://orcid.org/0000-0001-9974-4867

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
