## [Reviewer comments · BMJ Open]

ARTICLE DETAILS

TITLE (PROVISIONAL)	Appraising the quality standard of clinical practice guidelines related to central venous catheter-related thrombosis prevention: a systematic review of clinical practice guidelines
AUTHORS	Zhang, Jing; Wu, Yongya; Zhang, Shuai; Yao, Wenmo; Bu, Faqian; Wang, Aoxue; Hu, Xiuying; Wang, Guan

VERSION 1 – REVIEW

REVIEWER	Liqin, Ding Tianjin University of Traditional Chinese Medicine
REVIEW RETURNED	14-Jul-2023

GENERAL COMMENTS	This manuscript evaluated the quality and analyze the content of clinical practice guidelines regarding central venous catheter-related thrombosis to provide evidence for formulating an evidence-based practice guide and a risk assessment scale to prevent it. We believe that this manuscript is very meaningful and has high reference value for the development of future clinical guidelines for CVC. Thus, I suggest that this manuscript should be accepted after minor revision. Here are some of my comments: 1. Some of the references cited in this manuscript are earlier, and it is recommended to supplement recent publications.2. This manuscript lists the limitations of the guidelines in the discussion and analysis section of the recommended opinions, but doesn't provide corresponding suggestions for the future development of the guidelines. It is recommended to add suggestions for the designation of future guidelines in this section.3. This manuscript lists the basic characteristics of CRT prevention related guidelines in the discussion and analysis section of the recommended opinions of the guidelines. However, the author describes the characteristics in one paragraph, which is a little unclear. It is recommended that the author describe the characteristics of this section in points to better facilitate readers' understanding of the basic characteristics of CRT prevention related guidelines.4. There are still some Syntax errors in this manuscript. It is suggested that the author check the full text and modify the language of the manuscript.
--

REVIEWER	Estrada-Orozco, Kelly Universidad Nacional de Colombia, Health Technologies and Policies Assessment Group, School of Medicine
REVIEW RETURNED	04-Dec-2023

GENERAL COMMENTS	The topic of the study is important in the field of prevention of unwanted events associated with health care, so I consider that it may be important for its results to be known. The authors must make important changes in the wording of the document and in light of the use of methodological tools conclude appropriately on the aspects of quality of the guides and quality of the recommendations. Below are some suggestions in detail: The study includes a review of the literature to locate the CPGs as an initial step. Does this systematic review have an a priori protocol? If so, include this in the document. The document uses the AGREE 2 to evaluate the overall quality of the guide, which is methodologically correct, however, it is concluded about adequate quality in the majority of the documents, even in those with the domain of rigor in the minor elaboration of fifty%. It is striking that a document was not even clear in the systematic review or in the methods that led to the generation of recommendations; what would have to be reviewed is whether it really meets the criteria for good overall quality. I would suggest checking it out. In the scenario of evaluating clinical practice guidelines, the domains of rigor in the preparation and of editorial independence and conflict management are usually crucial to assign an AGREE 2 rating and even decide on the use or not of the CPGs. The evaluation of the recommendations was not carried out in a standardized way, there are currently instruments that help in these activities and that are globally accepted, such as the AGREE REX, this focuses on evaluating the quality and applicability of each recommendation, and can help with the comparability of the results, add validity to the evaluation process and even consider relevant aspects of the recommendations that help evaluate their quality and clarity. Results are presented in the discussion section, there is a qualitative synthesis with generation of analysis categories of the recommendations that are not presented as results but in discussion. It is suggested that the authors reorganize all the results in the corresponding section, so that they can discuss the results and their implications for clinical practice more broadly. Finally, it may help to have this document reviewed by an official translator to review the use of some English words that may make the results easier to understand.
--

REVIEWER	Ntalouka, Maria P General University Hospital of Larissa, Anaesthesiology
REVIEW RETURNED	06-Dec-2023

GENERAL COMMENTS	Congrats for the effort. Please revise carefully the whole manuscript for typos and details in English language. Otherwise a well presented and designed study
---

VERSION 1 – AUTHOR RESPONSE

Reviewer: 1

Dr. Ding Liqin, Tianjin University of Traditional Chinese Medicine

Comments to the Author:

This manuscript evaluated the quality and analyzed the content of clinical practice guidelines regarding central venous catheter-related thrombosis to provide evidence for formulating an evidence-based practice guide and a risk assessment scale to prevent it. We believe that this manuscript is very meaningful and has high reference value for the development of future clinical guidelines for CVC. Thus, I suggest that this manuscript should be accepted after minor revision.

Here are some of my comments:

-Some of the references cited in this manuscript are earlier, and it is recommended to supplement recent publications.

Re: Thank you for your comments, we have carefully checked and updated the references in this article, and the specific updated references are as follows:

7. Falanga A, Ay C, Di Nisio M, Gerotziakas G, Jara-Palomares L, Langer F, et al. Venous thromboembolism in cancer patients: ESMO Clinical Practice Guideline. *Ann Oncol.* 2023 May;34(5):452–67.

18. Duffett L. Deep Venous Thrombosis. *Ann Intern Med.* 2022 Sep;175(9): ITC129–44.

24. Brouwers MC, Kho ME, Browman GP, Burgers JS, Cluzeau F, Feder G, et al. AGREE II: Advancing guideline development, reporting, and evaluation in health care. *Preventive Medicine.* 2010 Nov 1;51(5):421–4.

25. Development and Validation of a Tool to Assess the Quality of Clinical Practice Guideline Recommendations. *JAMA Netw Open.* 2020;3(5):e205535. Published 2020 May 1. doi:10.1001/jamanetworkopen.2020.5535

-This manuscript lists the limitations of the guidelines in the discussion and analysis section of the recommended opinions but doesn't provide corresponding suggestions for the future development of the guidelines. It is recommended to add suggestions for the designation of future guidelines in this section.

Re: Thank you for your valuable comments, we have again analyzed the article in depth based on your comments, and provided corresponding suggestions for the future development of the guide in the discussion section of the article, which is supplemented as follows:

“In summary, the quality of the nine clinical guidelines included in this study is moderately high, but there are some differences in the guidelines in different regions and countries. The guidelines need to be further improved in the following aspects: (1) there is no risk assessment scale dedicated to CRT, and a targeted evaluation form should be developed based on evidence-based evidence, which is conducive to the accurate selection of appropriate preventive measures and anticoagulation treatments; (2) clinical practice guidelines and risk assessment scales related to CRT prevention based on evidence-based concepts should be formulated to enable patients to participate in the study based on the clinical status quo to increase the practicability of the guidelines.”

-This manuscript lists the basic characteristics of CRT prevention-related guidelines in the discussion and analysis section of the recommended opinions of the guidelines. However, the author describes the characteristics in one paragraph, which is a little unclear. It is recommended that the author describe the characteristics of this section in points to better facilitate readers' understanding of the basic characteristics of CRT prevention-related guidelines.

Re: Thank you for your valuable comments, we have consulted the relevant articles in PubMed based on your comments, carefully read the description of the basic characteristics of the guidelines related to the prevention of CRT in the articles, and modified our table according to the description of the basic characteristics of the guidelines. The specific changes are shown in the following table:

Table S2. Includes the basic characteristics table of the guideline

Year	Publishing agency	Subject	language	Whether or not evidence-based guidelines	Number of references
2020	Vascular Alliance China Chapter	Catheter-related thrombosis prevention	Chinese	Not	62
2019	American Society of Clinical Oncology	Prevention of cancer-related blood clots	English	Yes	155
2018	Chinese Medical Association	Diagnosis and prevention of thrombosis	Chinese	Not	41
2019	Spanish Society of Medical Oncology	Prevention and treatment of cancer-related thrombosis	English	Yes	88

-There are still some Syntax errors in this manuscript. It is suggested that the author check the full text and modify the language of the manuscript.

Re: We must apologize for the poor language of our manuscript. According to your suggestions, we have carefully checked and modified the whole manuscript, and we hope that the language level has been substantially improved.

Reviewer: 2

Dr. Kelly Estrada-Orozco, Universidad Nacional de Colombia, Center For Evidence to Implementation

Comments to the Author:

The topic of the study is important in the field of prevention of unwanted events associated with health care, so I consider that it may be important for its results to be known.

The authors must make important changes in the wording of the document and light of the use of methodological tools conclude appropriately on the aspects of the quality of the guides and quality of the recommendations. Below are some suggestions in detail:

- The study includes a review of the literature to locate the CPGs as an initial step. Does this systematic review have an a priori protocol? If so, include this in the document.

Re: Thank you very much for your suggestion, we prepared a priori protocol to guide this review, which has been uploaded as a supplementary document - priori protocol. We planned to register this review on PROSPERO; however, it only accepts the registration of the study of people

- The document uses AGREE 2 to evaluate the overall quality of the guide, which is methodologically correct, however, it is concluded that adequate quality in the majority of the documents, even in those with the domain of rigor in the minor elaboration of fifty%. It is striking that a document was not even clear in the systematic review or in the methods that led to the generation of recommendations; what would have to be reviewed is whether it meets the criteria for good overall quality. I would suggest checking it out. In the scenario of evaluating clinical practice guidelines, the domains of rigor in the preparation and of editorial independence and conflict management are usually crucial to assigning an AGREE 2 rating and even deciding on the use or not of the CPGs.

Re: Thank you very much for your suggestion, which is very important for this study. Based on your opinion, we consulted with statistical experts, re-understood all items of the scale, added a reviewer and the review team re-examined the 9 CPGs included in the study according to AGREE 2 and updated the evaluation results table in the article as follows:

Table S3 Specific standardized scores for each AGREE II domain

number	range and objective	Standardized score by domain (%)					The number of standardized scores \geq 60%.	recommend level
		Participants	Rigor	Clarity	Application	independence		
1	62.96	31.48	31.94	64.81	31.94	44.44	2	B
2	75.93	72.22	63.89	68.52	66.67	75.00	6	A
3	74.07	81.48	77.08	55.56	72.22	69.44	5	B
4	38.89	33.33	30.56	61.11	41.67	61.11	2	B
5	70.37	68.52	66.67	62.96	47.22	61.11	5	B
6	79.63	83.33	84.03	77.78	80.56	72.22	6	A
7	64.81	62.96	64.58	61.11	66.67	52.78	5	B
8	72.22	77.78	72.92	72.22	69.44	72.22	6	A

9	61.11	35.19	65.28	59.26	68.06	63.89	4	B
Mean Score	66.67	60.70	61.88	64.81	60.49	63.58	-	-

- The evaluation of the recommendations was not carried out in a standardized way, there are currently instruments that help in these activities and that are globally accepted, such as the AGREE REX, which focuses on evaluating the quality and applicability of each recommendation, and can help with the comparability of the results, add validity to the evaluation process and even consider relevant aspects of the recommendations that help evaluate their quality and clarity.

Re: Thank you for your valuable comments, which are of great significance for the improvement of the research validity and recognition of this study. Based on your recommendations, our research team used AGREE REX to evaluate the recommendations for inclusion in the guidelines in a standardized manner, making the evaluation process more effective and the evaluation results more acceptable. At the same time, we have added the specific evaluation process and evaluation results of AGREE REX in the article as a supplementary document and uploaded it as Table S4. The evaluation results are as follows:

Table S4 Specific standardized scores of AGREE REX in each field

number	AGREE REX Score by Sector (%)			Overall rating
	Clinical applicability	Values and preferences	implement ability	
1	42.59	41.67	50.00	43.83
2	61.11	61.11	66.67	62.35
3	74.07	65.28	77.78	70.99
4	37.04	41.67	61.11	44.44
5	50.00	61.11	72.22	59.88
6	83.33	73.61	77.78	77.78

7	59.26	50.00	63.89	56.17
8	66.67	66.67	77.78	69.14
9	48.15	59.72	52.78	54.32

- Results are presented in the discussion section, and there is a qualitative synthesis with the generation of analysis categories of the recommendations that are not presented as results but in the discussion. It is suggested that the authors reorganize all the results in the corresponding section so that they can discuss the results and their implications for clinical practice more broadly.

Re: Thanks for your suggestion, we have again carefully analyzed the data obtained from the scale scoring and explained it in detail in the results section of the manuscript, as follows:

Results of AGREE II and AGREE REX

evaluations The included guidelines used AGREE II and AGREE-REX to evaluate the quality of methodologies and recommendations, respectively. After scoring the 9 guidelines on the AGREE II scale, it was found that 3 guidelines were recommended as A and 6 were recommended as B, as shown in Table 2. The average score in all areas was greater than 60%. The ICC of the scores of the three reviewers for each domain of each guideline was greater than 0.8, indicating that the agreement among the reviewers was good. Specifically, Area 1 ("Scope and Purpose") addresses the general scope and purpose of CPGS, the specific clinical problems that need to be addressed, and the target population. The average score was 66.67%, of which 8 articles scored more than 60% and 5 papers scored more than 70%. Area 2 ("Participants") reflects whether the normative process includes input and participation from relevant stakeholders. The average score was 60.70%, and 6 of the articles scored above the average score. In terms of "rigor" (area 3), the average score was 61.88%, with 7 guidelines scoring above the average. Guideline 4 has the lowest scores in the three areas of scope

and purpose, who is involved, and rigor, while most of the rest describe the development process, albeit with varying levels of detail.

Area 4 ("Clarity") focuses on the specificity and clarity of these items, including the clear articulation of options for health management and the making of key recommendations that are easy to identify. With an average score of 64.81%, there was the smallest difference in all guideline scores in this area, even though guideline scores 1, 2, 8, and 9 were all below the average. "Application" (area 5) refers to facilitators and obstacles to the implementation of recommendations within the context of the guidelines, strategies for their implementation, and possible resource implications. The CPGS scored an average of 60.49 % in this area, with 3 scores below 60%. Finally, the sixth area ("independence") aims to ensure that there is a lack of bias in the development of guidelines. The score in this area was relatively good, with a median CPGS of 63.58%, 3 above 70%, and 7 above 60%. Most described funding, potential impact, or associated conflicts of interest. The guidelines scored slightly worse than the others in the areas of participants, rigor of development, applicability, and editorial independence.

After evaluating the nine guidelines according to the AGREE REX scale, the overall scores of guidelines 3 and 6 were > 70%, which was of high quality, and the overall scores of the other guidelines were \geq 30%, which was of moderate quality. The three reviewers agreed well, with an ICC value greater than 0.8, and each guideline scored slightly lower than the implement ability domain in the values and preference domains and clinical practicality. In the field of clinical applicability, guideline 6 scored the highest score of 83.33%, with a large score difference between guidelines. guideline 1 had the lowest score of 42.59%, as detailed in Table S4.

Comparative analysis of the specific content of CRT preventive measures in the guidelines

After reading all the recommendations and precautions included in the guidelines, it was found that the content of CRT prevention can be mainly divided into three aspects: risk assessment, preventive measures, and health education, and the specific content and methods are shown in Table 5. Due to differences in the topic of the guidelines and the participants of the study, the focus and level of detail of the prevention strategies recommended for CRT vary among the guidelines. Therefore, for analysis and comparison, the relevant contents are summarized in Table S6.

-Finally, it may help to have this document reviewed by an official translator to review the use of some English words that may make the results easier to understand.

Re: Thank you for your comments, we have invited professionals to carefully check and revise the word use and grammar of this study, and hope that the revised content will make the results of this study more conducive to readers.

Reviewer: 3

Dr. Maria P Ntalouka, General University Hospital of Larissa

Comments to the Author:

Congrats on the effort.

Please revise carefully the whole manuscript for typos and details in English language.

Otherwise a well presented and designed study

Re: Thank you for your valuable suggestions, we have carefully read checked, and revised the spelling and grammar of the words in this study, and hope that this study will be more conducive to readers.

VERSION 2 – REVIEW

REVIEWER	Estrada-Orozco, Kelly Universidad Nacional de Colombia, Health Technologies and Policies Assessment Group, School of Medicine
REVIEW RETURNED	25-Jan-2024
GENERAL COMMENTS	Most of suggestions were incorporated by the authors!